# Transfer Phenomena of Nanoliposomes by Live Imaging of Primary Cultures of Cortical Neurons

**DOI:** 10.3390/pharmaceutics14102172

**Published:** 2022-10-12

**Authors:** Elodie Passeri, Philippe Bun, Kamil Elkhoury, Michel Linder, Catherine Malaplate, Frances T. Yen, Elmira Arab-Tehrany

**Affiliations:** 1LIBio Laboratory, University of Lorraine, 54505 Vandoeuvre-lès-Nancy, France; 2Qualivie Team, UR AFPA Laboratory, University of Lorraine, 54505 Vandoeuvre-lès-Nancy, France; 3NEURIMAG, Institute of Psychiatry and Neuroscience of Paris (IPNP)—Inserm U1266, University of Paris, 75014 Paris, France

**Keywords:** nanoliposomes, cortical neurons, endocytosis, transfer phenomena, single particle trafficking

## Abstract

Soft nanoparticles, and in particular, nanoliposomes (NL), have attracted increasing interest for their use in food, nutraceuticals, and in particular, in pharmaceutics for drug delivery. Recent data using salmon lecithin NL suggest that these NL, rich in omega-3 (n-3) fatty acids, can improve the bioavailability and transport of molecules through the blood brain barrier (BBB) to target the brain for the prevention and treatment of neurodegenerative diseases. The objective of this study was to characterize the physicochemical properties and analyze the transfer phenomena of salmon lecithin NL over time in neurons to better understand the behavior of NL in an intracellular environment. To test this, primary cultures of cortical neurons from rat embryos were incubated with salmon lecithin NL from day 3 after cell culture, for up to 104 h. The physicochemical properties of NL such as size, speed, morphology and the diffusion coefficient in the live cultures, were studied over time. Image analysis of cell morphology showed dendritic growth and neuronal arborization after 48 h of exposure to NL, for up to 104 h. Results showed an NL stability in size, speed and diffusion coefficient over time, with a peak at 48 h, and then a return to baseline value at the end of incubation. The average speed and diffusion coefficient achieved provided important information on the mode of entry of NL into neurons, and on the slow diffusion rate of NL into the cells. Analysis of videos from 2 h to 104 h showed that significant levels of NL were already internalized by neurons after 3 h incubation. NL appearance and intracellular distribution indicated that they were packed in intracellular compartments similar to endocytic vesicles, suggesting internalization by an active endocytic-like process. The results obtained here demonstrate internalization of NL by cortical neurons by an active endocytic-like process, and suggest the potential use of NL for time-release of therapeutics aimed towards prevention or treatment of neurodegenerative diseases.

## 1. Introduction

Liposomes were discovered in 1965 by A.D. Bangham when he observed that phospholipids formed closed lipid bilayer vesicles when surrounded by an aqueous medium [1]. They can be defined as closed, bilayered structures composed mainly of lipid and/or phospholipid molecules [2], ranging in size between 20 and ∼1000 nm to several µm in diameter depending on preparation methodology and lipid composition [3]. Phospholipids, amphiphilic molecules, are the main components of NL, which contain a water soluble and hydrophilic head section and a hydrophobic tail section [3].

NL are the nanometric-sized liposomes. They are colloidal closed structures, formed by small lipid bilayers dispersed in an aqueous compartment [4]. NL of salmon lecithin are rich in n-3 polyunsaturated fatty acids (PUFA), crucial components of neuronal membranes, particularly in terms of fluidity. By virtue of the high content of n-3 PUFA, their unique phospholipid bilayer structure shows strong similarity to the neuronal membrane physiology in the brain [5]. NL can serve two roles. First, they themselves are bioactive, being composed of n-3 PUFA such as docosahexaenoic (DHA) and eicosapentaenoic (EPA) acids. NL can also serve as vectors or carriers of both hydrophilic and/or hydrophobic bioactive molecules or drugs, protecting molecules from the external environment [5]. NL, therefore, represent an advantageous and innovative nanotechnology for the encapsulation, drug delivery and targeting of bioactive compounds [6]. NL can be modified for specific tissue or cell targeting and can be used to protect therapeutic agents from premature degradation. By virtue of the NL origin, they represent a biodegradable and biocompatible delivery system in vivo. NL can be used for transport to the brain [7], thus, improving bioavailability and transport of molecules through the BBB to reach relevant brain regions to treat neurodegenerative diseases [8,9,10].

Recently, liposomes have attracted increasing interest, and there have been many studies in different fields [11] including food, nutraceuticals, [12] and pharmaceutics [5] in order to better understand their preparation and characterization as well as developing their use as carriers of drugs delivery and vaccines and other bioactive materials [3,5,13]. 

Recent results have demonstrated the neurotrophic effects of NL in an in cellulo model as well as their bioavailability in vivo [14,15]. Nevertheless, little information is available on NL interaction with neurons and the mechanisms of internalization involved at the cellular level.

Fluorescence microscopy imaging techniques such as single particle tracking (SPT) are used to observe the movement and trajectory of particles in living cells and to observe intracellular traffic [16]. Highly sensitive fluorescence microscopy techniques make it possible to follow individual nanoparticles as they are absorbed into living cells with high temporal and spatial resolution. The type of deviation of molecules is widely studied in the field of biophysics. From the analysis of the trajectories, the movement and the average transport speed as well as the diffusion coefficient can be determined. Such an analysis, therefore, provides important information regarding the absorption pathway and localization of nanoparticles [17]. Here, we used this technology to better understand the movement of NL on the cell surface and to determine the type of molecular diffusion that was involved [18]. Indeed, diffusion is fundamental to most cellular processes, providing important clues to biological systems and pharmaceutical applications of nanotransporter systems [19].

A green extraction technique for the preparation of n-3 PUFA-rich NL from salmon lecithin obtained from salmon head by-products has been developed in our laboratory. We recently demonstrated, for the first time, the bioavailability of NL PUFA in neuronal cell culture and mouse models, with a higher enrichment in n-3 PUFA after administration of NL at the cerebral level, in particular in the cortex, compared with the peripheral tissues (liver). In addition, these naturally-derived NL have been shown to be biocompatible in vivo [15].

We propose that salmon lecithin NL could be used as both a drug carrier and a supplement for the delivery of n-3 rich PUFA to the brain, thereby maintaining neuronal function and synaptic plasticity, preventing cognitive decline related to aging and reducing the risk of neurodegenerative diseases. For this purpose, a better understanding is required of NL interaction at the cellular level: how do NL interact with neurons and by what cellular mechanism are they internalized?

The aim of this study was to characterize the physicochemical properties and the diffusion coefficient of salmon lecithin NL in primary cortical neurons over time using SPT, in order to better understand the behavior of NL within neuronal cells.

## 2. Materials and Methods

### 2.1. Lecithin Extraction

Lecithin was extracted enzymatically and purified from *Salmo salar* head through an enzymatic process using low temperature enzymatic hydrolysis without the need of organic solvents [20,21]. 

### 2.2. Fatty Acid Composition Analysis 

Fatty acid methyl esters (FAMEs) were prepared as previously described [22]. FAMEs separation was accomplished on gas chromatography (Perichrom, Saulx-lès-Chartreux, France). The temperatures of the injector and detector were set at 250 °C. Column temperature was initially programmed at 120 °C for 3 min, then it was raised to 180 °C at a rate of 2 °C min^−1^ and increased at 220 °C for 25 min. Fatty acids were determined using Standard mixtures (Supelco, Sigma-Aldrich, Bellefonte, PA, U.S.A.). Runs were conducted in triplicate.

### 2.3. Lipid Classes of Nanoliposomes 

Lipidic classes of salmon phospholipids were analyzed using a Iatroscan MK-5 TLC-FID (Iatron Laboratories Inc., Tokyo, Japan) as described previously [23]. 

Two migrations were performed to determine the proportion of neutral and polar lipid fractions. The area percentage of each pic were expressed as the mean of three repetitions.

### 2.4. Preparation of Nanoliposomes

To create NL, salmon lecithin was prepared at a concentration of 2% (*w/w*) in distilled water. The solution was capped under nitrogen to prevent lipid oxidation and agitated for 4 h. Then the mixture was sonicated (Sonicator Vibra-cell 75115, 500 W, Bioblock Scientific Co.) at 40 kHz at 40% of full power for 360 s (1 s on, and 1 s off). NL samples were stored in the dark at 4 °C in glass bottles [24,25,26,27].

### 2.5. Nanoliposomes Characterization 

Dynamic light scattering (DLS) using a Malvern Zetasizer Nano ZS (Malvern Instruments Ltd., Malvern, U.K.) was used to measure the average hydrodynamic particle diameter (Hd), polydispersity index (PDI), and ζ-potential of NL. NL samples were diluted (with ultrapure distilled water, ratio of 1:200) Measurements were performed with a scattering angle of 173° at 25 °C, a refractive index (RI) of 1.471, and absorbance of 0.01. The measurements were conducted in standard capillary electrophoresis cells equipped with gold electrodes (DTS 1070). For each condition, three independent measurements were obtained.

### 2.6. Transmission Electron Microscopy

NL structures were visualized using TEM with a negative staining method [28]. To reduce the concentration of NL, the samples were diluted 25-fold with ultrapure distilled water. To stain the samples, the same volumes of the diluted solution were mixed with an aqueous solution of ammonium molybdate (2%). Then, the samples were left at room temperature for 3 min and incubated for 5 min on a copper mesh coated with carbon. After drying, samples were examined using a Philips CM20 TEM (Philips, Dresden, Germany) associated with an Olympus TEM CCD camera to determine the morphology of the NL.

### 2.7. Primary Cultures of Cortical Neurons

Primary cultures of rat embryo cortical neurons were prepared as described previously [29]. Pregnant female Wistar rats at day 17 of gestation were sacrificed by gas anesthesia (isoflurane), and the embryos were removed. The animals were housed in a certified animal facility (approval code no. C75-14-03). All experiments involving rats were performed in accordance with Directive 2010/63/EU of the European Parliament and of the Council of 22 September 2010, on the protection of animals used for scientific purposes. The experimental protocol used was approved by the ethic committee for animal experimentation of Paris Descartes (2018–2019).

Embryos cortices were collected for enzymatic digestion using trypsin/EDTA in MSS medium. Then cells were plated at 10 × 10^4^ cells/cm² in Petri dishes pre-coated with poly-l-ornithine (15 µg/mL, Sigma). The cells were cultured in neuronal culture medium M2 containing serum-free DMEM-F12 medium (Invitrogen, Illkirch, France) containing 0.5 µM insulin, 60 µM putrescine, 30 nM sodium selenite, 100 µM transferrin, 10 nM progesterone, and 0.1% (*w/v*) ovalbumin (Sigma).

The cell cultures were maintained in an incubator at 37 °C in a humidified 5% CO_2_ atmosphere. Day (D) 0 was the day of the preparation of the primary culture; NL were added on D3 at a concentration of 10 µg/mL. Cytotoxicity of NL on cells was previously determined while using the MTT assay [14].

### 2.8. Microscopy Imaging of Live Cortical Neurons 

A microscope LEICA Spinning disk X1 (Leica Microsystems, Mannheim, Germany). was used for all the acquisitions. The objective used was a 63x (NA 1.3) glycerol-immersion. The camera was a Hamamatsu Orca Flash 4.0v3 (Hamamatsu Photonics, Shizuoka, Japan). The wavelength of the laser used to image was 488 nm laser and 405 nm laser, the exposure time of the camera was between 100 and 200 ms. The acquisition frequency was 5 s, and the total duration of the films acquired under the microscope was 300 s. The films were captured for 300 s with one frame every 5 s, for 60 frames. Time lapse imaging was performed at 37 °C using a spinning-disk microscope.

### 2.9. Image Analysis of Nanoliposomes in Cortical Neurons 

First, the films were processed in Fiji-Image J, pixel: 6.5/63 (=0.1031 µm), format 1024 × 1024. Then, ICY software version 2.4.0.0 was used for analyses of the NL movements to estimate the average speed, size, the mean square displacement and coefficient diffusion of the NL. The NL were detected using Spot Detector plugin. In order to avoid detecting background noise, the average size of the detected objects was around 2 pixels. Trajectories whose total displacement was not greater than 10 pixels (=1 µm) were not tracked, the same was true for instantaneous displacements of at least 1 pixel.

Once the detection occurred, the Spot Tracking plugin was used to extract the trajectories of the detected NL. Within this plugin, filters were applied to filter the trajectories of the NL according to several criteria: the instantaneous displacement, the total displacement, the number of frames where the NLs are present (frame interval: 10 frames, total displacement 1 µm, net displacement 0.1 µm). These filters allow to have optimal measurement of the trajectories. Then, from these tracking groups, for each time presented, the analysis focused on 30 NL which were tracked for the time of the film (300 s), and we extracted the average speed, the area, the mean square displacement (MSD).

For the translational global contribution to the trajectories, the cellular movements during the acquisition time were negligible or even non-existent, which did not lead to correcting the movement due to the cell.

### 2.10. Mean Square Displacement and Diffusion Coefficient

Analysis of particle trajectories within the cell included the quantification of this evolution in time and in space and was linked to the criteria of the MSD [16]. One of the main purposes of MSD analysis is the extraction of the diffusion coefficient value D, and the type of particle diffusion [30]. The MSD < r^2^ > of diffusing particle was obtained as:(1)$$\langle r^{2}\rangle =4{D\Delta t}^{\alpha }$$
where D is the diffusion coefficient, t the time and α the exponent.

To determine the types of NL motion, the MSD was calculated and their curves were analyzed depending on the exponent α. At each time point, for all of the individual trajectories identified, MSD was calculated with the average of all the intervals [16]. Each interval had a duration of 5 s. This averaging yielded values for short lags but was found to be very dispersed and with a small number of values for long lags. We, therefore, centered on the intervals between 0 and 20 in order to have a correct statistical representation. The diffusion coefficient was extracted from the MSD formula (1).

### 2.11. Statistical Analysis

All values were expressed as means ± SD. Results were analyzed by one-way analysis of variance (ANOVA) followed by Tukey’s comparison test to assess differences between means. Statistical significance was determined, *p* < 0.05 was considered to be significant.

## 3. Results and Discussion

### 3.1. Fatty Acid Composition of Salmon Lecithin

Main fatty acid (FA) composition in salmon lecithin was analyzed by gas chromatography and results are shown in Table 1. Analyses revealed that 41.35% of total FA were PUFAs, of which n-3 PUFAs were the most abundant FAs with a total percentage of 29.13%. The ratio of n-6/n-3 ratio was 0.42. The most significant proportions of n-3 PUFAs were DHA (18.04%) and EPA (7.55 %). Salmon lecithin also contained 31.09% of monounsaturated FAs (MUFAs), particularly oleic acid (C18:1n-9, 26.16%) and 27.56% of saturated FAs (SFAs), which are mainly represented by palmitic acid (C16:0, 20.08%).

### 3.2. Lipid Classes of Salmon Lecithin

An Iatroscan analysis was utilized to determine lipid classes. Results showed that salmon lecithin was composed primarily of phospholipids (67.65 ± 0.90%), then triglycerides (31.20 ± 0.40%) and a low level of total cholesterol (1.15 ± 0.10%), (*n* = 10). Phosphatidylcholine (PC) was the most abundant class of total phospholipids (42%) [15]. Hence, although salmon lecithin was found to be composed primarily of phospholipids, it contains neutral lipids and a low level of total cholesterol.

### 3.3. Physicochemical Characterization of Nanoliposomes after Their Preparation

NL were prepared using probe-sonication from a 2% (*w/v*) salmon lecithin solution (Figure 1A). The polydispersity index (PDI), ζ-potential, and average particle size of NL were measured immediately after their preparation (Figure 1B). The PDI of NL was found to be 0.25 ± 0.003 which indicates a controlled size distribution and a narrow dispersity of the NL population. ζ-potential measurements revealed that the NL particle’s surface electrical charge was found to be highly negative (−52.4 ± 0.7 mV). This was indicative of the stability of the colloidal dispersions [31]. The particle mean size was found to be 103.5 ± 1.5 nm (*n* = 3). These NL contain multiple bilayers and the NL’s size was further confirmed via transmission electron microscopy (TEM) that also confirmed their spherical morphology (Figure 1C), as previously shown [14,15,32,33].

### 3.4. Diameter and Average Speed of Nanoliposomes over Time after Incubation with Cortical Neurons

Primary culture of embryo cortical neurons was prepared, followed by addition of NL containing a fluorescent probe on Day 3. Cells were monitored over time during incubation at 37 °C in the presence of 10 µg/mL NL. Supplementation of NL in culture media at this concentration was previously shown to accelerate neuronal development, neurite outgrowth and synaptogenesis in embryonic cortical neurons [14].

Analysis of the evolution of the size of the nanoliposomes, smaller than 1000 nm, over time indicated that there was no significant difference in size between the start and the end of the experimental period. Indeed, NL diameter remained stable between 2 h and 104 h of incubation (Table 2).

Several factors influence the internalization of a particle [34,35]. Size and stability are the most important properties in a nanocarrier such as liposomes because it can affect the efficiency of drug delivery. The size stability problem is more crucial for nanoparticles than for other drug delivery systems because nanoparticles have a large specific surface area [36]. These data demonstrate the stability of salmon lecithin-derived NL incubated in the presence of cortical neurons.

The average speed over time gradually increased from 2 h to reach its maximum average speed at 48 h. It then decreased to values close to the baseline at 104 h. Statistically significant differences were found at different time points and in particular between those measured at 2 h and at 48 h (*p* < 0.001).

No significant correlation was observed between average speed and diameter (*r* = 0.31, n.s.).

The NL presented a stability in size and speed over time, and there was not a significant difference between the first and last times. Particle size and particle size distribution does have an impact on drug delivery. Nanocomposite formulations with a constant and narrow size distribution represent an advantage and are needed to achieve optimal results [34].

In addition to the speed, size, and morphology, one of the important characteristics of the particle is the type of diffusion of the particle within the living cell.

### 3.5. Mean Square Displacement and Diffusion Coefficient

Analyses of the MSD average and the diffusion coefficient were performed on six time frames, namely, 2 h, 7 h, 26 h, 48 h, 72 h and 100 h. After detection, tracking of particles, and calculation of the MSD, their curves were analyzed as a function of the exponent α, and the motion of NL was characterized over different times (Figure 2A). The calculation of the MSD represents measurement of the average area explored by the particle over time in relation to a starting position.

The different MSD curves over time showed α values of less than 1, ranging between 0.30 (2 h) and 0.77 (7 h). Trajectories were classified according to α and found to correspond to an anomalous diffusion (subdiffusion) with an α value of <1 [37,38,39]. Interestingly, the 2 h and 100 h time curves appeared to be more representative of a confined diffusion (Figure 2B).

**Figure 2 pharmaceutics-14-02172-f002:**
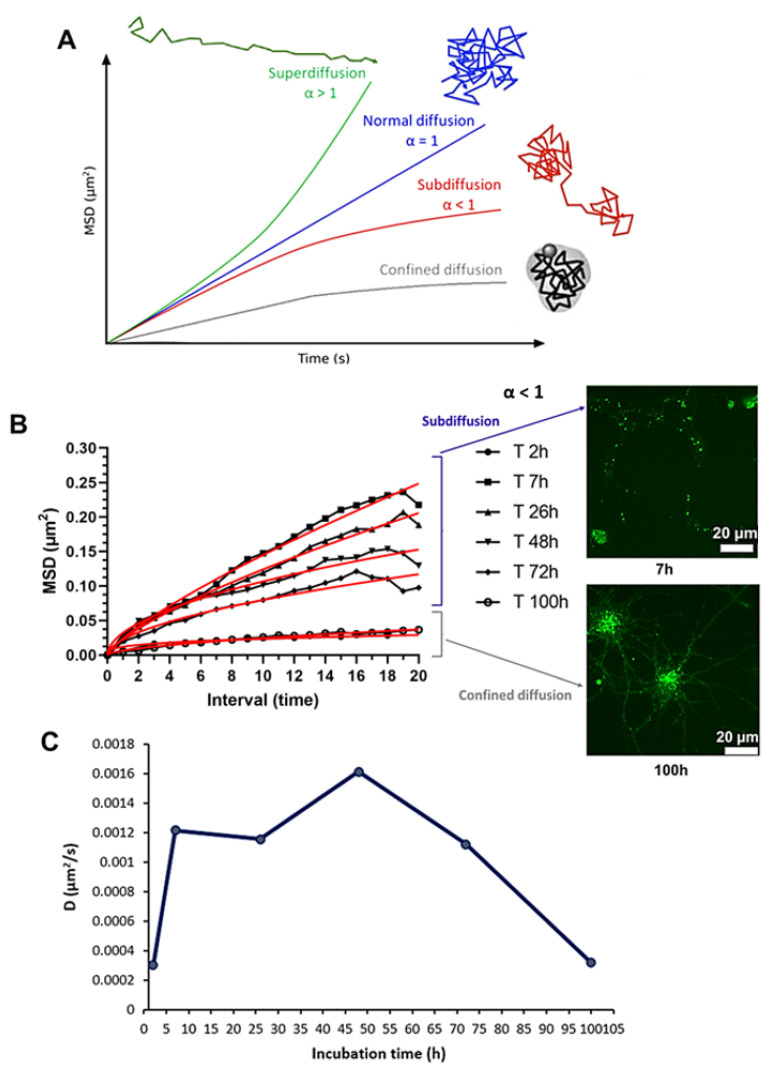
Mean square displacement (MSD). (**A**) MSD: type of diffusion and particle trajectory. The time dependence of the MSD characterizes the diffusional behavior of particles: a linear line (green) indicates normal diffusion (α = 1), a directed motion (blue) represents a superdiffusion (α > 1). When the MSD increases more slowly than for normal diffusion (α < 1), the particle can present a subdiffusion (red) or a confined diffusion (gray) (Adapted from Kapanidis et al. [40] which is licensed under a Creative Commons Attribution-(CC BY 4.0) International License). (**B**) MSD average curves of nanoliposomes at 2, 7, 26, 48, 72 and 100 h of incubation and representative fluorescence images of primary cortical neurons treated with NL at 7 and 100 h of incubation. NL labeled with ATTO 488 (green) (scale bar: 20 μm) are shown on the right to illustrate the subdiffusion (7 h, top) and confined diffusion (100 h, bottom). The mathematical adjustments (red curve) were obtained by applying formula 1 (Section 2.10) to the raw data. (**C**) Diffusion coefficient of nanoliposomes at 2, 7, 26, 48, 72 and 100 h of incubation.

This result was consistent with other studies. In fact, single particle tracking analyses showed anomalous diffusion (subdiffusion) including nanoparticles, in living cells [41,42] and within the cell membranes [18,43], which can be explained by non-Brownian motion, such as obstructed diffusion imposed by molecular obstacle, by almost stationary obstacles such as membrane domains, and/or of traps involving a distribution of the binding energies [44,45]. Another possible mechanism giving rise to this subdiffusion is that particles take long rests between periods of free scattering motion [46,47,48,49]. Indeed, subdiffusion slows long-distance diffusional exchanges but it proves to be beneficial for local interactions in cells [50].

The diffusion coefficient was maximum at 48 h of incubation (D = 0.00161µm^2^/s) and returned to its starting value (D = 0.000320 µm^2^/s) at 100 h (Figure 2C). This corresponded to the same time frame as for the average speed (peak at 48 h), where the average speed and the diffusion coefficient over time were strongly correlated (*r* = 0.97, *p* < 0.01).

The average speed, the diffusion coefficient and the subdiffusive motion are consistent with internalization of NL via an active endocytic-like process [51]. The slower diffusion observed could have been due to the packing of NL in endocytic vesicles, and the intracellular transport of NL between the neuron cell body and neurites.

The phenomenon of diffusion depends on many factors such as speed, size, temperature, environmental conditions, and intrinsic properties of the particle. Our results, therefore, suggest that the dynamics of anomalous diffusion of NL within cortical neurons could be explained by interactions between particles and spatial obstacles such crowding and obstruction effects [39] which slow down diffusion in the long term, which could be explained by the accumulation of the NL in endocytic vesicles and could lead to subdiffusive behavior [51].

### 3.6. Internalization of Nanoliposomes Inside Cortical Neurons

Analysis of videos from 2 h to 104 h showed that NL were detected intracellularly in neurons after 3 h of treatment (Figure 3A). Small amounts of NL were already detectable under a microscope from 2 h; these levels were increased in the neuronal cell images at 3 h. NL were in a cell culture system, their low mobility at 2 h could be explained by the size of nanoliposome and their multilayer properties.

### 3.7. Image Analysis of Cortical Neurons Morphology after Incubation over Time

Image analysis of cell morphology showed high levels of dendritic growth and neuronal arborization after 48 h of exposure to NL, consistent with our previous study showing neurotrophic effects of NL on cortical neurons [14,32,33]. Arborization was present even after 72 and 100 h of treatment (Figure 3B), and up to 104 h.

### 3.8. Fluorescence Image Analysis of Nanoliposomes over Time

Tracking over time revealed that these vesicles migrated from the extensions to the cell bodies between 7 and 48 h. Up to 48 h, the NL appeared to be present in dendrites. In the 48 h and 72 h images, the vesicles containing fluorescent NL were localized primarily in the cell bodies (Figure 3B). After 100 h, we observed that the vesicles located at the cell body seemed to fuse together into larger structures while, at the same time, smaller vesicles were redirected towards the neuronal extensions, suggesting endocytic trafficking of the NL for several hours after their internalization. Based on the appearance and distribution of the fluorescent structures observed, this suggested that the NL were internalized by endocytosis.

Endocytosis is the process of capturing a substance or particle from outside the cell by enveloping it in the cell membrane and transporting it into the cell. The endosomal recycling network represents a dynamic system for sorting and re-exporting or degrading internalized membrane components. Internalized vesicles can undergo fusion to form early endosomes [52].

More precisely, the internalized cargo can be sorted and recycled to the plasma membrane via the early endosome, sent to the trans-Golgi network via retrograde traffic mechanisms, or routed via the late endosome to the lysosome for degradation [53,54,55].

The internalization of particles by neurons occurs via two main endocytic pathways, macropinocytosis and receptor-mediated endocytosis (RME), the main ones of which are clathrin-mediated endocytosis (CME) forming endocytic clathrin-coated vesicles, with a size of 70 to 150–200 nm, and caveolin-mediated endocytosis (CVME), with vesicle size ranging from 60 to 80 nm [56,57]. It is generally accepted that the internalization of nanoparticles in non-phagocytic cells is greater for smaller particles. However, studies reported that particle sizes up to 150 nm are predominantly internalized via CME or CVME with a maximum size of 200 nm [58,59]. There are other endocytic mechanisms, such as RhoA-, ARF6-, flotillin- or CDC42-mediated endocytosis, but they do not contribute significantly to the uptake of cellular nanoparticles, compared with RME.

Further investigation is required to determine the specific endocytic mechanisms involved in NL cellular uptake and internalization, and to determine NL localization in the different intracellular compartments of neurons, more precisely.

## 4. Conclusions

The results obtained here using real-time single particle tracking demonstrated internalization of the NL by cortical neurons after 3 h of incubation and indicated that NL are packed in intracellular compartments similar to endocytic vesicles. NL remained stable with regards to size and subdiffusion phenomena, consistent with intracellular transport in endocytic vesicles, pointing towards the potential use of NL for time-release of therapeutics aimed towards prevention or treatment of neurodegenerative diseases [60].

## Figures and Tables

**Figure 1 pharmaceutics-14-02172-f001:**
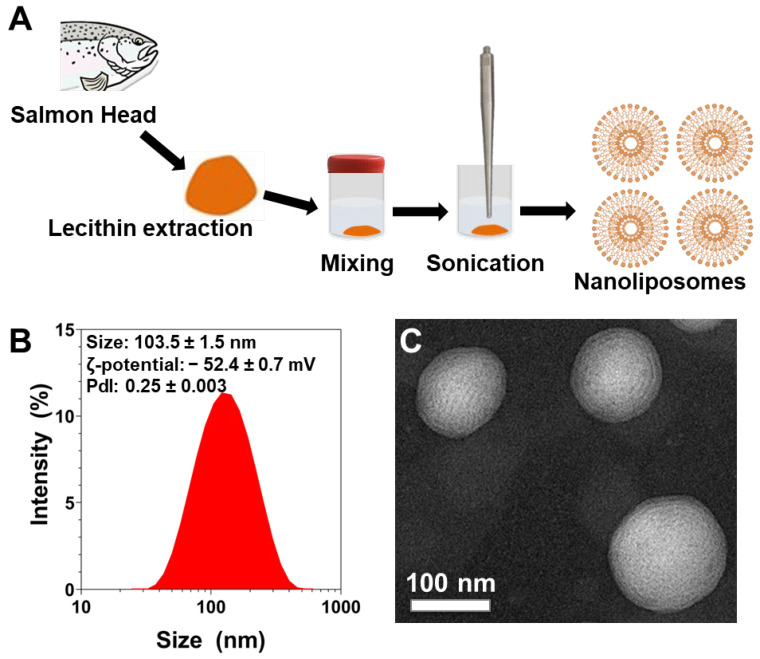
(**A**) Schematic representation of the fabrication process of salmon-derived nanoliposomes. (**B**) Size distribution measurements, and average size, ζ-potential, and polydispersity index values of liposomes measured by dynamic light scattering. The reported data are represented as mean ± SD of at least three individual experiments. (**C**) TEM image of nanoliposomes after their preparation.

**Figure 3 pharmaceutics-14-02172-f003:**
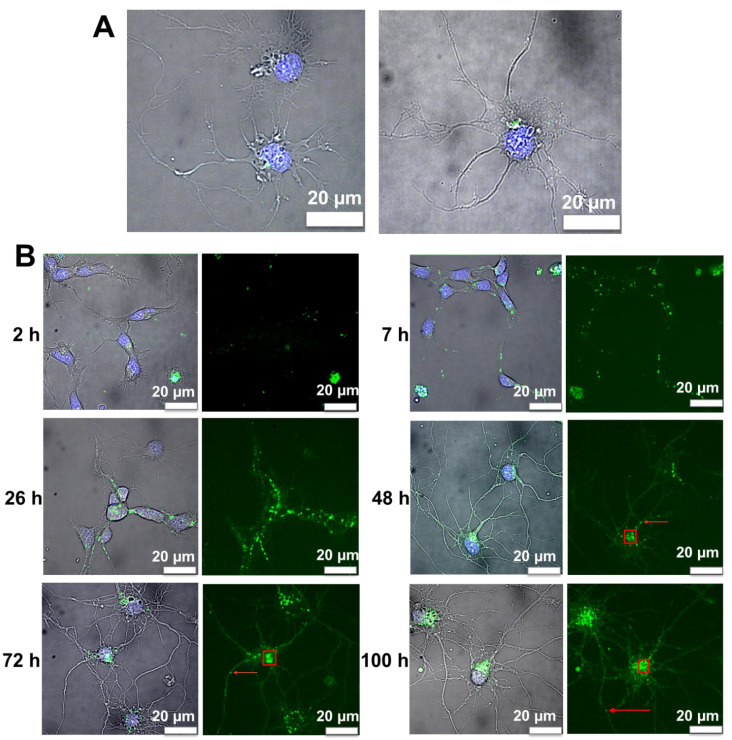
Representative images of primary cortical neurons treated with NL. (**A**) Representative images of NL after 3 h of incubation (merge) showing NL labeled with ATTO 488 (green) and DAPI-labeled nuclei (blue); scale bar: 20 μm. (**B**) Representative images of primary cortical neurons treated with NL labeled with ATTO 488 (green) at 2, 7, 26, 48, 72 and 100 h of incubation. Nuclei are labeled with DAPI (blue) (scale bar: 20 μm). The left panels of each time point are merge images to show both DAPI and ATTO 488 labels. The red boxes and arrows represent examples of cell bodies and extensions/dendrites, respectively.

**Table 1 pharmaceutics-14-02172-t001:** Fatty acid composition of nanoliposomes.

Fatty Acids (% Total)	Salmon Lecithin
Myristic acid (C14:0)	1.67 ± 0.02
Pentadecanoic acid (C15:0)	0.42 ± 0.01
Palmitic acid (C16:0)	20.08 ± 0.11
Heptadecanoic acid (C17:0)	0.41 ± 0.03
Stearic acid (C18:0)	4.98 ± 0.04
**Saturated Fatty Acids**	**27.56**
Palmitoleic acid (C16:1n-7)	1.74 ± 0.01
Oleic acid (C18:1n-9)	26.16 ± 0.21
Vaccenic acid (C18:1n-7)	2.56 ± 0.04
Eicosenoic acid (C20:1n-9)	0.63 ± 0.02
**Monounsaturated Fatty Acids**	**31.09**
Linoleic acid (C18:2n-6)	6.98 ± 0.11
Gamma-linolenic acid (C18:3n-6)	3.13 ± 0.05
Arachidonic acid (C20:4n-6)	2.11 ± 0.03
n-6 total	12.22
Alpha-linolenic acid (C18:3n-3)	1.63 ± 0.07
Eicosapentaenoic acid (C20:5n-3)	7.55 ± 0.05
Docosapentaenoic acid (C22:5n-3)	1.91 ± 0.08
Docosahexaenoic acid (C22:6n-3)	18.04 ± 0.09
n-3 total	29.13
**Polyunsaturated Fatty Acids**	**41.35**
DHA/EPA	2.39
n-6/n-3	0.42

The results presented are mean ± standard deviation (SD) of % total fatty acids (*n* = 3).

**Table 2 pharmaceutics-14-02172-t002:** Diameter and average speed of nanoliposomes over time from 2 to 104 h.

Incubation Time	Diameter (μm)	Avg. Speed (μm/s)
2 h	0.312 ± 0.044	0.007 ± 0.004 ^c,d,e,f,g,h,i,j,k,l,m,n,o^
3 h	0.317 ± 0.046	0.012 ± 0.002 ^d,e,f,g,h,i,j,k,l,m^
5 h	0.317 ± 0.058	0.017 ± 0.007 ^a,h,i^
6 h	0.339 ± 0.039	0.022 ± 0.006 ^a,b,o,p,q^
7 h	0.315 ± 0.042	0.023 ± 0.005 ^a,b,i,o,p,q^
8 h	0.290 ± 0.054	0.024 ± 0.003 ^a,b,i,o,p,q^
26 h	0.325 ± 0.042	0.024 ± 0.005 ^a,b,n,o,p,q^
34 h	0.335 ± 0.055	0.027 ± 0.004 ^a,b,c,m,n,o,p,q^
48 h	0.329 ± 0.037	0.029 ± 0.006 ^a,b,c,e,k,l,m,n,o,p,q^
50 h	0.314 ± 0.050	0.024 ± 0.007 ^a,b,n,o,p,q^
55 h	0.303 ± 0.051	0.022 ± 0.005 ^a,b,i,o,p,q^
72 h	0.333 ± 0.041	0.020 ± 0.005 ^a,b,i,o,p,q^
76 h	0.330 ± 0.042	0.019 ± 0.006 ^a,b,h,i,o,p,q^
80 h	0.295 ± 0.048	0.018 ± 0.006 ^a,g,h,i,jo,p,q^
96 h	0.294 ± 0.051	0.013 ± 0.005 ^a,d,e,f,g,h,i,jk,l,m,n^
100 h	0.324 ± 0.046	0.012 ± 0.004 ^d,e,f,g,h,i,j,kl,m,n^
104 h	0.300 ± 0.059	0.012 ± 0.004 ^d,e,f,g,h,i,j,k,l,m^

The results were expressed as the mean ± standard deviation (SD), (*n* = 30). Statistical differences (*p* < 0.05) are shown for diameter and for average, as compared for each incubation time with 2 h ^a^, 3 h ^b^, 5 h ^c^, 6 h ^d^, 7 h ^e^, 8 h ^f^, 26 h ^g^, 34 h ^h^, 48 h ^i^, 50 h ^j^, 55 h ^k^, 72 h ^l^, 76 h ^m^, 80 h ^n^, 96 h ^o^, 100 h ^p^, 104 h ^q^.

## Data Availability

Not applicable.

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
