# Peer review of "Transfer Phenomena of Nanoliposomes by Live Imaging of Primary Cultures of Cortical Neurons"

_pharmaceutics, 2022, doi:10.3390/pharmaceutics14102172_

Round 1
Reviewer 1 Report
The paper “Transfer phenomena of nanoliposomes by live imaging of primary cultures of cortical neurons” reports the characterization of liposomes obtained from salmon lecithin. Lipids composition of the extract is well determined and the structural properties of the liposomes are measured. The cytotoxicity, internalization of NL are studied. In particular authors aim to obtain a measure of the dynamical and structural properties of the NL into rat embryo cortical neurons. From results they interpret the internalization as guided by endocytosis.
The paper is appealing and faces biological issues under a biotechnological perspective, with a biophysical approach.
However, before being considered for publication, some major points must be addressed or better explained.
The authors should better explain how they can obtain a MSD every second, having, as they say, a collection for 300s of 60 frames, captured every 5sec.
If msd is given every sec, one needs several displacement of the particles in 1 sec, in order to calculate a msd(s).
Why not represent msd in log-log plot? The analysis is not clear. It should pass from the origin. The analysis represented by the red curves is not well explained.
Could they better explain the meaning of this sentence? “Filters were applied (frame interval: 10 frames, total displacement 1 μm, net displacement 0.1 μm)”
The average speed calculated is correlated with the msd and is changing over time, thus why the authors affirm that there is stability in NL speed.
How was determined the average speed? 30 particles were observed for how long? I suggest to show the speed distribution for the beginning and for 48h.
The model in the picture resembles to unilamellar vesicles, while from TEM the NL appear as multilamellar, so the graphical view should be changed and it is misleading.
How the authors explain the low mobility of NL very early at t=2h? Is it due to an internalization process? Where are localized at this step the NL? Time course images reported in Fig3 represent different cells, the 2h image is poor of NL. One could expect that NL are in the extracellular space, but this is not evident in the image and this is against the low mobility. Could you give a deeper evidence of the average results? Representative trajectories could be shown. Could you comment on this?
Since the particle is detected by accounting 2 pixels and movements are very small, authors should comment on their experimental resolution by compared to the statistical errors presented and also if and how they consider any translational global contribution to the trajectories.
The authors should better explain why the distribution and evolution observed of the NL support the endocytosis. It is not clear to me. Some reference linking the diffusion coefficient values to the interpretation of endocytosis should be reported. Both references [41, 42] reported to sustain the interpretation are simulative and it would be nice have a comparison with other experimental diffusion coefficient values obtained from similar experimental analysis.

Author Response
The paper “Transfer phenomena of nanoliposomes by live imaging of primary cultures of cortical neurons” reports the characterization of liposomes obtained from salmon lecithin. Lipids composition of the extract is well determined and the structural properties of the liposomes are measured. The cytotoxicity, internalization of NL are studied. In particular authors aim to obtain a measure of the dynamical and structural properties of the NL into rat embryo cortical neurons. From results they interpret the internalization as guided by endocytosis.
The paper is appealing and faces biological issues under a biotechnological perspective, with a biophysical approach.
However, before being considered for publication, some major points must be addressed or better explained.
The authors should better explain how they can obtain a MSD every second, having, as they say, a collection for 300s of 60 frames, captured every 5sec.
If msd is given every sec, one needs several displacement of the particles in 1 sec, in order to calculate a msd(s).
We thank the reviewer for these comments. We have changed the unit in the curve. It is indeed not a second but an interval which has a duration of 5 seconds. We have made the corrections in the text and in Figure 2B.
Why not represent msd in log-log plot? The analysis is not clear. It should pass from the origin. The analysis represented by the red curves is not well explained.
We have not represented the msd of Figure 2B in log-log plot to remain comparable to that of Figure 2A.
The curve in Figure 2B has been modified to pass from the origin.
The mathematical adjustments (red curve) were obtained by applying formula 1 (section 2.1, equation (1) to the raw data.
Could they better explain the meaning of this sentence? “Filters were applied (frame interval: 10 frames, total displacement 1 μm, net displacement 0.1 μm)”
The filters applied were used to filter the trajectories of the NL according to several criteria: the instantaneous displacement, the total displacement, the number of frames where the NLs are present. The edit has been incorporated into the text.
The average speed calculated is correlated with the msd and is changing over time, thus why the authors affirm that there is stability in NL speed.
There is a stability of the NL speed over time because we do not see a significant difference between the first and the last times.
How was determined the average speed? 30 particles were observed for how long? I suggest to show the speed distribution for the beginning and for 48h.
The average speed was determined with the spot tracking plugin of ICY software. For each time presented, the analysis focused on 30 NL who were tracked the time of the film (300s) and we extracted the average speed (see Figure 1 below).
Figure 1. Average speed of nanoliposomes at 2, 3, 5, 5, 7, 8, 26, 34 and 48 hours of incubation.
The results ​​were expressed as the mean ± standard deviation (SD), (n=30)
The model in the picture resembles to unilamellar vesicles, while from TEM the NL appear as multilamellar, so the graphical view should be changed and it is misleading.
These NL contain multiple bilayers. The graphical abstract and the figure 1A have been changed to show that NL as such.
How the authors explain the low mobility of NL very early at t=2h? Is it due to an internalization process? Where are localized at this step the NL? Time course images reported in Fig3 represent different cells, the 2h image is poor of NL. One could expect that NL are in the extracellular space, but this is not evident in the image and this is against the low mobility. Could you give a deeper evidence of the average results? Representative trajectories could be shown. Could you comment on this?
The nanoliposomes are in cell culture system. The low mobility can be explained by the size of nanoliposome and their multilayers properties.
Since the particle is detected by accounting 2 pixels and movements are very small, authors should comment on their experimental resolution by compared to the statistical errors presented and also if and how they consider any translational global contribution to the trajectories.
These filters make it possible to be very restrictive on the trajectories (lines 182-183). Trajectories whose total displacement is not greater than 10 pixels (=1 um) are not tracked; the same is true for instantaneous displacements of at least 1 pixel.
For the translational global contribution to the trajectories, the cellular movements during the acquisition time are negligible or even non-existent, which did not lead to correcting the movement due to the cell.
The authors should better explain why the distribution and evolution observed of the NL support the endocytosis. It is not clear to me. Some reference linking the diffusion coefficient values to the interpretation of endocytosis should be reported. Both references [41, 42] reported to sustain the interpretation are simulative and it would be nice have a comparison with other experimental diffusion coefficient values obtained from similar experimental analysis
We have added references. In continuation of this study, further investigation is needed to determine the specific endocytic mechanisms involved in NL cellular uptake and internalization into endocytic vesicles.
We thank the reviewer for your attention. We have made the corrections requested into the text and we hope it now makes everything clear.

Reviewer 2 Report
Transfer phenomena of nanoliposomes by live imaging of primary cultures of cortical neurons by Elodie Passeri et al.
This study characterizes salmon lecithin nanoliposomes (NL) transfer to rat primary cortical neurons and describes NL physicochemical properties (size, speed, morphology and the diffusion coefficient). The authors conclude that neuronal internalization of NL is an active endocytic-like process and they suggest the potential use of NL for therapeutics in neurodegenerative diseases.
Minor edits, comments and suggestions
Line 23: “…and the diffusion coefficient in the live cultures were studied over time.”
Line 25: “Results showed a NL stability in size, speed and diffusion coefficient…”
Line 50: “…shows strong similarity to the cell membrane physiology in the brain [5].” Please be more specific. Which cell type in the brain?
Line 129: “…a refractive index (RI) at 1471, and absorbance…”
Line 152: “The cell cultures were maintained …”
Line 156: “2.8 Microscopy imaging of live cortical neurons”
Line 166: “Then, the videos of NL movement were carried out in the ICY software were analyzed to estimate their average speed, size, the mean square displacement and coefficient diffusion. ICY software was used for analyses.” This is confusing, please be more clear.
Line 173: What is MSD? Please write full name also.
Line 185: “…for short lags but was found to be very dispersed and with…”
Line 207: “…to determine lipids classes.”
Line 231: “…after incubation with cortical neurons” The title can be condensed because the other details are described in the next line.
Line 233: “…embryonic cortical neurons…”
Line 236: “…to accelerate neuronal development…”
Please indicate statistical significance in Table 2.
Fig 2B: Please comment on why could 2h and 100 h show the same confined diffusion?
Line 286: “In fact, SPT analyses…” Please write SPT full name.
Line 324: Fig 3 Legend: “The red boxes represent the cell bodies and the arrows the extensions/dendrites. The red boxes and arrows represent examples of cell bodies and extensions/dendrites, respectively.” These sentences are redundant. There is only one arrow, please place more arrows.
Line 347: “…endocytosis (CME) forming endocytic clathrin-coated vesicles…”
Line 354: “…compared to RME.”
Line 358: “..real-time single particle tracking”
For a more accurate NL localization in neurons, specific markers for endosomes (like EEA1), lysosomes (lysotracker), etc. can also be used in addition of NL labeling and DAPI. Please comment on this briefly.
Line 24, Abstract: “Image analysis of cell morphology showed dendritic growth and neuronal arborization after 48 h of exposure to NL, and for up to 104 h.” Please quantify from the images this change in neurite growth after NL exposure as it will give valuable information on the NL effect on neuronal well-being. This may be relevant to the future objectives of potentially using NL for prevention and treatment of neurodegenerative diseases.
Author Response
This study characterizes salmon lecithin nanoliposomes (NL) transfer to rat primary cortical neurons and describes NL physicochemical properties (size, speed, morphology and the diffusion coefficient). The authors conclude that neuronal internalization of NL is an active endocytic-like process and they suggest the potential use of NL for therapeutics in neurodegenerative diseases.
Minor edits, comments and suggestions
Line 23: “…and the diffusion coefficient in the live cultures were studied over time.” The change has been incorporated into the text.
Line 25: “Results showed a NL stability in size, speed and diffusion coefficient…” The change has been incorporated into the text.
Line 50: “…shows strong similarity to the cell membrane physiology in the brain [5].” Please be more specific. Which cell type in the brain? It is the neuronal cell in the brain. The edit has been incorporated into the text
Line 129: “…a refractive index (RI) at 1471, and absorbance…” The change has been incorporated into the text.
Line 152: “The cell cultures were maintained …” The change has been incorporated into the text.
Line 156: “2.8 Microscopy imaging of live cortical neurons” The change has been incorporated into the text.
Line 166: “Then, the videos of NL movement were carried out in the ICY software were analyzed to estimate their average speed, size, the mean square displacement and coefficient diffusion. ICY software was used for analyses.” The edit has been incorporated into the text. We hope it now makes clear.
Line 173: What is MSD? Please write full name also. The change has been incorporated into the text.
Line 185: “…for short lags but was found to be very dispersed and with…” The change has been incorporated into the text.
Line 207: “…to determine lipids classes.” The change has been incorporated into the text.
Line 231: “…after incubation with cortical neurons” The title can be condensed because the other details are described in the next line. The change has been incorporated into the text.
Line 233: “…embryonic cortical neurons…” The change has been incorporated into the text.
Line 236: “…to accelerate neuronal development…” The change has been incorporated into the text.
Please indicate statistical significance in Table 2. . We have made the edits requested in Table 2 and in the legend.
Fig 2B: Please comment on why could 2h and 100 h show the same confined diffusion?
In Figure 2B, the 2h and 100h incubation times show the same confined diffusion. This type of motion was determined from the MSD. The MSD was calculated and their curves were analyzed as a function of the exponent α.
Line 286: “In fact, SPT analyses…” Please write SPT full name. The change has been incorporated into the text.
Line 324: Fig 3 Legend: “The red boxes represent the cell bodies and the arrows the extensions/dendrites. The red boxes and arrows represent examples of cell bodies and extensions/dendrites, respectively.” These sentences are redundant. There is only one arrow, please place more arrows.
We have made the edits requested in the figure 3 and the change has been incorporated into the legend.
Line 347: “…endocytosis (CME) forming endocytic clathrin-coated vesicles…” The change has been incorporated into the text.
Line 354: “…compared to RME.” The change has been incorporated into the text.
Line 358: “..real-time single particle tracking” The change has been incorporated into the text.
For a more accurate NL localization in neurons, specific markers for endosomes (like EEA1), lysosomes (lysotracker), etc. can also be used in addition of NL labeling and DAPI. Please comment on this briefly.
Further investigation is required to determine the specific endocytic mechanisms involved in NL cellular uptake and internalization and to determine more precisely NL localization in the different intracellular compartments of neurons. And the authors thank the reviewer for this recommendation. It would be interesting for the future study to use specific markers for endosomes and lysosomes for more precise NL localization in neurons.
Line 24, Abstract: “Image analysis of cell morphology showed dendritic growth and neuronal arborization after 48 h of exposure to NL, and for up to 104 h.” Please quantify from the images this change in neurite growth after NL exposure as it will give valuable information on the NL effect on neuronal well-being. This may be relevant to the future objectives of potentially using NL for prevention and treatment of neurodegenerative diseases.
Quantitation of neurite outgrowth after NL exposure is very difficult, and requires a specialized software. We have previously demonstrated neurotrophic effects of NL in this same cell culture model, and provided analyses and images of neurite outgrowth, synaptogenesis, and arborization (Malaplate et al, Marine Drugs, 17(7), 406, 2019).
We thank the reviewer for the feedback. We have made the corrections requested. The changes have been incorporated into the text, the figure, the table and legends.

Round 2
Reviewer 1 Report
It is fine with me with the improvement done